# Fundamental Limits of Crystalline Equivariant Graph Neural Networks: A Circuit Complexity Perspective

## Abstract

Graph neural networks (GNNs) have become a core paradigm for learning on relational data. In materials science, equivariant GNNs (EGNNs) have emerged as a compelling backbone for crystalline-structure prediction, owing to their ability to respect Euclidean symmetries and periodic boundary conditions. Despite strong empirical performance, their expressive power in periodic, symmetry-constrained settings remains poorly understood. This work characterizes the intrinsic computational and expressive limits of EGNNs for crystalline-structure prediction through a circuit-complexity lens. We analyze the computations carried out by EGNN layers acting on node features, atomic coordinates, and lattice matrices, and prove that, under polynomial precision, embedding width $d = O(n)$ for $n$ nodes, $O(1)$ layers, and $O(1)$-depth, $O(n)$-width MLP instantiations of the message/update/readout maps, these models admit a simulation by a *uniform* $\mathsf{TC}^0$ threshold-circuit family of polynomial size (with an explicit constant-depth bound). Situating EGNNs within $\mathsf{TC}^0$ provides a concrete ceiling on the decision and prediction problems solvable by such architectures under realistic resource constraints and clarifies which architectural modifications (e.g., increased depth, richer geometric primitives, or wider layers) are required to transcend this regime. The analysis complements Weisfeiler-Lehman style results that do not directly transfer to periodic crystals, and offers a complexity-theoretic foundation for symmetry-aware graph learning on crystalline systems.

## 1 Introduction

Graphs are a natural language for relational data, capturing entities and their interactions in domains ranging from molecules and materials (Merchant et al., 2023) to social (Sankar et al., 2021) and recommendation networks (Ying et al., 2018). Graph neural networks (GNNs) have consequently become a standard tool for learning on such data: the message-passing paradigm aggregates information over local neighborhoods to produce expressive node and graph representations that power tasks such as node/edge prediction and graph classification. This message-passing template (i.e., graph convolution followed by nonlinear updates) underlies many successful architectures and applications (Jumper et al., 2021; Brehmer et al., 2023).

Recently, *equivariant* graph neural networks (EGNNs) (Satorras et al., 2021) have emerged as a promising direction for modeling crystalline structures in materials science. By respecting Euclidean symmetries and periodic boundary conditions, EGNNs encode physically meaningful inductive biases, enabling accurate predictions of structures, energies, and related materials properties directly from atomic coordinates and lattice parameters (Schmidt et al., 2022; Merchant et al., 2023). In practice, E(3)/E($n$)-equivariant message passing and related architectures achieve strong performance while avoiding some of the computational burdens of higher-order spherical-harmonics pipelines (Thomas et al., 2018; Liao & Smidt, 2022), and they have been adapted to periodic crystals (Jiao et al., 2023; AI4Science et al., 2023). Moreover, EGNN-style backbones are now widely used within crystalline generative models, including diffusion/flow-based approaches that model positions, lattices, and atom types jointly (Jiao et al., 2023; Yang et al., 2023; Zeni et al., 2023).

Despite this progress, fundamental questions about *expressive power* remain. In particular, we ask:

> *What are the intrinsic computational and expressive limits of EGNNs for crystalline-structure prediction?*

Prior theory for (non-equivariant) message-passing GNNs analyzes expressiveness through the lens of the Weisfeiler–Lehman (WL) hierarchy (Xu et al., 2018; Morris et al., 2019; 2020), establishing that standard GNNs are at most as powerful as 1-WL and exploring routes beyond via higher-order or subgraph-based designs (Morris et al., 2019; Maron et al., 2019; Cotta et al., 2021; Qian et al., 2022); other lines study neural models via circuit-complexity bounds. However, WL-style results focus on discrete graph isomorphism and typically abstract away continuous coordinates and symmetry constraints, while most existing circuit-complexity analyses target different architectures (e.g., Transformers (Li et al., 2024b; Chen et al., 2025a)). These differences make such results ill-suited to crystalline settings, where periodic lattices, continuous 3D coordinates, and E($n$)-equivariance are first-class modeling constraints. This motivates a tailored treatment of EGNNs for crystals.

In this paper, we investigate the *fundamental expressive limits of EGNNs in crystalline-structure prediction* (Kaba & Ravanbakhsh, 2022; Jiao et al., 2023; Miller et al., 2024). Rather than comparing against WL tests, we follow a circuit-complexity route (Chiang, 2024; Liu, 2025): we characterize the computations performed by EGNN layers acting on node features, atomic coordinates, and lattice matrices, and we quantify the resources required to simulate these computations with uniform threshold circuits. Placing EGNNs within a concrete circuit class yields immediate implications for the families of decision or prediction problems such models can (and provably cannot) solve under realistic architectural and precision constraints. This perspective complements WL-style analyses and is naturally aligned with architectures, such as EGNNs, that couple graph structure with continuous, symmetry-aware geometric features.

Our contributions are summarized as follows:

- **Formalizing EGNNs' structure.** We formalize the definition of EGNNs (Definition 3.10).
- **Circuit-complexity upper bound for EGNNs.** Under polynomial precision, embedding width $d = O(n)$, $O(1)$ layers, and $O(n)$-width $O(1)$-depth MLP instantiations of the message/update/readout maps, we prove that the EGNN class in Definition 3.10 can be simulated by a *uniform* $\mathsf{TC}^0$ circuit family (Theorem 5).

**Roadmap.** In Section 2, we review the relevant works. In Section 3, we show the basic concepts and notations. In Section 4, we analyze the circuit complexity of components. In Section 5, we present our main results. Finally, in Section 6, we conclude our work.

## 2 RELATED WORK

**CSP and DNG in Materials Discovery** Early methods for CSP and DNG approached materials discovery by generating a large pool of candidate structures and then screening them with high-throughput quantum mechanical calculations (Kohn & Sham, 1965) to estimate stability. Candidates were typically constructed through simple substitution rules (Wang et al., 2021) or explored with genetic algorithms (Glass et al., 2006; Pickard & Needs, 2011). Later, machine learning models were introduced to accelerate this process by predicting energies directly (Schmidt et al., 2022; Merchant et al., 2023).

To avoid brute-force search, generative approaches have been proposed to directly design materials (Court et al., 2020; Yang et al., 2021; Nouira et al., 2018). Among them, diffusion models have gained particular attention, initially focusing on atomic positions while predicting the lattice with a variational autoencoder (Xie et al., 2021), and more recently modeling positions, lattices, and atom types jointly (Jiao et al., 2023; Yang et al., 2023; Zeni et al., 2023). Other recent advances incorporate symmetry information such as space groups (AI4Science et al., 2023; Jiao et al., 2024; Cao et al., 2024), leverage large language models (Flam-Shepherd & Aspuru-Guzik, 2023; Gruver et al., 2024), or employ normalizing flows (Wirnsberger et al., 2022).

**Flow Matching for Crystalline Structures** Flow Matching (Lipman et al., 2023; Tong et al., 2023b; Dao et al., 2023) has recently established itself as a powerful paradigm for generative modeling, showing remarkable progress across multiple areas. The initial motivation came from addressing the heavy computational cost of Continuous Normalizing Flows (CNFs) (Chen et al., 2018),

as earlier methods often relied on inefficient simulation strategies (Rozen et al., 2021; Ben-Hamu et al., 2022). This challenge inspired a new class of Flow Matching techniques (Albergo & Vanden-Eijnden, 2022; Tong et al., 2023a; Heitz et al., 2023), which learn continuous flows directly without resorting to simulation, thereby achieving much better flexibility. Thanks to its straightforward formulation and strong empirical performance, Flow Matching has been widely adopted in large-scale generation tasks. For instance, (Davtyan et al., 2023) proposes a latent flow matching approach for video prediction that achieves strong results with far less computation. (Zhang et al., 2025a) applies consistency flow matching to robotic manipulation, enabling efficient and fast policy generation. (Jing et al., 2024) develops a flow-based generative model for protein structures that improves conformational diversity and flexibility while retaining high accuracy. (Luo et al., 2024) introduces CrystalFlow, a flow-based model for efficient crystal structure generation. Overall, Flow Matching has proven to be an efficient tool for generative modeling across diverse modalities. *Notably, EGNN-style backbones have become a de facto choice for crystalline structure generative modeling: diffusion- and flow-based pipelines pair symmetry-aware message passing with periodic boundary handling to jointly model positions, lattices, and compositions* (Jiao et al., 2023; Yang et al., 2023; Zeni et al., 2023; Luo et al., 2024). In these systems, the equivariant message-passing core supplies an inductive bias that improves sample validity and stability while reducing reliance on higher-order tensor features (Satorras et al., 2021; AI4Science et al., 2023; Jiao et al., 2024).

**Geometric Deep Learning.** Geometric deep learning, particularly geometrically equivariant Graph Neural Networks (GNNs) that ensure E(3) symmetry, has achieved notable success in chemistry, biology, and physics (Jumper et al., 2021; Bronstein et al., 2021; Brehmer et al., 2023; Merchant et al., 2023; Qiu et al., 2023; Zhang et al., 2025b). In particular, equivariant GNNs have demonstrated superior performance in modeling 3D structures (Chanussot et al., 2021; Tran et al., 2023). Existing geometric deep learning approaches can be broadly categorized into four types: (1) Invariant methods, which extract features stable under transformations, such as pairwise distances and torsion angles (Schütt et al., 2018; Gasteiger et al., 2020; 2021); (2) Spherical harmonics-based models, which leverage irreducible representations to process data equivariantly (Thomas et al., 2018; Liao & Smidt, 2022); (3) Branch-encoding methods, encoding coordinates and node features separately and interacting through coordinate norms (Jing et al., 2020; Satorras et al., 2021); (4) Frame averaging frameworks, which model coordinates in multiple PCA-derived frames and achieve equivariance by averaging the representations (Puny et al., 2021; Duval et al., 2023).

While these architectures have pushed the boundaries of modeling geometric data in 3D structures, and advanced equivariant and invariant neural architectures in learning geometric data in chemistry, biology, and physics domains, the fundamental limitations of such architectures in crystalline structures still remain less explored. In this paper, we reveal the fundamental expressive capability limitation of equivariant GNNs via the lens of circuit complexity.

**Circuit Complexity and Machine Learning.** Circuit complexity is a fundamental notion in theoretical computer science, providing a hierarchy of Boolean circuits with different gate types and computational resources (Vollmer, 1999; Arora & Barak, 2009). This framework has recently been widely used to analyze the expressiveness of machine learning models: a model that can be simulated by a weaker circuit class may fail on tasks requiring stronger classes. A central line of work applies circuit complexity to understand Transformer expressivity. Early studies analyzed two simplified theoretical models of Transformers: SoftMax-Attention Transformers (SMATs) and Average-Head Attention Transformers (AHATs) (Liu et al., 2023; Merrill et al., 2022; Merrill & Sabharwal, 2023). Subsequent results have extended these analyses to richer Transformer variants, including those with Chain-of-Thought (CoT) reasoning (Feng et al., 2023; Li et al., 2024b; Merrill & Sabharwal, 2024), looped architectures (Giannou et al., 2023; Luca & Fountoulakis, 2024; Saunshi et al., 2025), and Rotary Position Embeddings (RoPE) (Chen et al., 2025a; Yang et al., 2025; Chen et al., 2025b). Beyond Transformers, circuit complexity has also been applied to other architectures such as state space models (SSMs) (Chen et al., 2025c), Hopfield networks (Li et al., 2024a), diffusion models (Cao et al., 2025; Chen et al., 2025d; Ke et al., 2025), and graph neural networks (GNNs) (Grohe, 2023; Cui et al., 2024; Li et al., 2025). In this work, we study the circuit complexity bounds of equivariant GNNs on crystalline structures, providing the first analysis of this kind.

## 3 PRELIMINARY

We begin by introducing some basics of crystal representations in Section 3.1, and then introduce the background knowledge of equivariant graph neural networks (EGNNs) in Section 3.2. Next, we present the basics in circuit complexity in Section 3.3.

### 3.1 REPRESENTATION OF CRYSTAL STRUCTURES

The unit cell representation describes the basis vectors of the unit cell, and all the atoms in a unit cell.

**Definition 3.1** (Unit cell representation of a crystal structure, implicit in page 3 of (Jiao et al., 2023)). *Let $A := [a_1, a_2, \ldots, a_n] \in \mathbb{R}^{h \times n}$ denote the list of description vectors for each atom in the unit cell. Let $X := [x_1, x_2, \ldots, x_n] \in \mathbb{R}^{3 \times n}$ denote the list of Cartesian coordinates of each atom in the unit cell. Let $L := [l_1, l_2, l_3] \in \mathbb{R}^{3 \times 3}$ denote the lattice matrix, where $l_1, l_2, l_3$ are linearly independent. The unit cell representation of a crystal structure can be defined as a triplet $\mathcal{C} := (A, X, L)$.*

The atom set representation describes a set containing an infinite number of atoms in the periodic crystal structure.

**Definition 3.2** (Atom set representation of a crystal structure, implicit in page 3 of (Jiao et al., 2023)). *Let $\mathcal{C} := (A, X, L)$ be a unit cell representation of crystal structure as Definition 3.1, where $A := [a_1, a_2, \ldots, a_n] \in \mathbb{R}^{h \times n}$, $X := [x_1, x_2, \ldots, x_n] \in \mathbb{R}^{3 \times n}$, and $L := [l_1, l_2, l_3] \in \mathbb{R}^{3 \times 3}$. The atom set representation of $\mathcal{C}$ is defined as follows:*

$$S(\mathcal{C}) := \{(a, x) : a = a_i, x = x_i + Lk, \forall i \in [n], \forall k \in \mathbb{Z}^3\},$$

*where $k$ is a length-3 column integer vector.*

**Definition 3.3** (Fractional coordinate matrix, implicit in page 3 of (Jiao et al., 2023)). *Let $\mathcal{C} := (A, X, L)$ be a unit cell representation of crystal structure as Definition 3.1, where $A := [a_1, a_2, \ldots, a_n] \in \mathbb{R}^{h \times n}$, $X := [x_1, x_2, \ldots, x_n] \in \mathbb{R}^{3 \times n}$, and $L := [l_1, l_2, l_3] \in \mathbb{R}^{3 \times 3}$. We say that $F := [f_1, f_2, \ldots, f_n] \in [0, 1)^{3 \times n}$ is a fractional coordinate matrix for $\mathcal{C}$ if and only if for all $i \in [n]$, we have:*

$$x_i = Lf_i.$$

**Definition 3.4** (Fractional unit cell representation of a crystal structure, implicit in page 3 of (Jiao et al., 2023)). *Let $\mathcal{C} := (A, X, L)$ be a unit cell representation of crystal structure as Definition 3.1. Let $F$ be a fractional coordinate matrix as Definition 3.3. The fractional unit cell representation of $\mathcal{C}$ is a triplet $\mathcal{C}_{\mathrm{frac}} := (A, F, L)$.*

**Fact 3.5** (Equivalence of unit cell representations, informal version of Fact A.1). *For any fractional unit cell representation $\mathcal{C}_{\mathrm{frac}} := (A, F, L)$ as Definition 3.4, there exists a unique corresponding non-fractional unit cell representation $\mathcal{C} := (A, X, L)$ as definition 3.1.*

Therefore, since both unit cell representations are equivalent, we only use the fractional unit cell representation in this paper. For notation simplicity, we may abuse the notation $\mathcal{C}$ to denote $\mathcal{C}_{\mathrm{frac}}$ in the following parts of this paper.

**Definition 3.6** (Fractional atom set representation of a crystal structure, implicit in page 3 of (Miller et al., 2024)). *Let $\mathcal{C}_{\mathrm{frac}} := (A, F, L)$ be a fractional unit cell representation of a crystal structure as Definition 3.4, where $A := [a_1, a_2, \ldots, a_n] \in \mathbb{R}^{h \times n}$, $F := [f_1, f_2, \ldots, f_n] \in \mathbb{R}^{3 \times n}$, and $L := [l_1, l_2, l_3] \in \mathbb{R}^{3 \times 3}$. The atom set representation of $\mathcal{C}$ is defined as follows:*

$$S_{\mathrm{frac}}(\mathcal{C}) := \{(a, f) : a = a_i, f = f_i + k, \forall i \in [n], \forall k \in \mathbb{Z}^3\},$$

*where $k$ is a length-3 column integer vector.*

### 3.2 EQUIVARIANT GRAPH NEURAL NETWORK ARCHITECTURE

We first define a useful transformation that computes the distance feature between each two atoms.

**Definition 3.7** ($k$-order Fourier transform of relative fractional coordinates). *Let $x \in (-1, 1)^3$ be a length-3 column vector. Without loss of generality, we let $k \in \mathbb{Z}_+$ be a positive even number.*

*Let the output of the $k$-order Fourier fractional coordinates be a matrix $Y \in \mathbb{R}^{3 \times k}$ such that $Y :=$ $\psi_{\mathrm{FT},k}(x)$. For all $i \in [3], j \in [k]$, each element of $Y$ is defined as:*

$$Y_{i,j} := \begin{cases} \sin(\pi j x_i), & j \text{ is even;} \\ \cos(\pi j x_i), & j \text{ is odd.} \end{cases}$$

Then, we define a single layer for the Equivariant Graph Neural Network (EGNN) on the fractional unit cell representation of crystals.

**Definition 3.8** (Pairwise Message)**.** *Let $\mathcal{C} := (A, F, L)$ be a fractional unit cell representation as Definition 3.4, where $A \in \mathbb{R}^{h \times n}$, $F := [f_1, f_2, \ldots, f_n] \in \mathbb{R}^{3 \times n}$, and $L \in \mathbb{R}^{3 \times 3}$. Let $H :=$ $[h_1, h_2, \ldots, h_n] \in \mathbb{R}^{d \times n}$ be a hidden neural representation for all the atoms. Let $\psi_{\mathrm{FT},k}$ be a $k$-order Fourier transform of relative fractional coordinates as Definition 3.7. Let $\phi_{\mathrm{msg}} : \mathbb{R}^d \times \mathbb{R}^d \times$ $\mathbb{R}^{3 \times 3} \times \mathbb{R}^{3 \times k} \to \mathbb{R}^d$ be an arbitrary function. We define the message $\mathsf{MSG}_{i,j}(F, L, H) \in \mathbb{R}^d$ between the $i$-th atom and the $j$-th atom for all $i, j \in [n]$ as follows:*

$$\mathsf{MSG}_{i,j}(F, L, H) := \phi_{\mathrm{msg}}(h_i, h_j, L^\top L, \psi_{\mathrm{FT},k}(f_i - f_j)).$$

**Definition 3.9** (One EGNN layer)**.** *Let $\mathcal{C} := (A, F, L)$ be a fractional unit cell representation as Definition 3.4, where $A := [a_1, a_2, \ldots, a_n] \in \mathbb{R}^{h \times n}$, $F := [f_1, f_2, \ldots, f_n] \in \mathbb{R}^{3 \times n}$, and $L := [l_1, l_2, l_3] \in \mathbb{R}^{3 \times 3}$. Let $H := [h_1, h_2, \ldots, h_n] \in \mathbb{R}^{d \times n}$ be a hidden neural representation for all the atoms. Let $\phi_{\mathrm{upd}} : \mathbb{R}^d \times \mathbb{R}^d \to \mathbb{R}^d$ be an arbitrary function. Let $\mathsf{MSG}$ be the message function defined as Definition 3.8. Let the output of the $i$-th EGNN layer $\mathsf{EGNN}_i(A, F, L, H)$ be a matrix $Y = [y_1, y_2, \ldots, y_n] \in \mathbb{R}^{d \times n}$, i.e., $Y := \mathsf{EGNN}_i(F, L, H)$. For all $i \in [n]$, each column of $Y$ is defined as:*

$$y_i := h_i + \phi_{\mathrm{upd}}(h_i, \sum_{j=1}^n \mathsf{MSG}_{i,j}(F, L, H)).$$

**Definition 3.10** (EGNN)**.** *Let $\mathcal{C} := (A, F, L)$ be a fractional unit cell representation as Definition 3.4, where $A \in \mathbb{R}^{h \times n}$, $F \in \mathbb{R}^{3 \times n}$, and $L \in \mathbb{R}^{3 \times 3}$. Let $q$ be the number of $\mathsf{EGNN}$ layers. Let $\phi_{\mathrm{in}} : \mathbb{R}^{h \times n} \to \mathbb{R}^{d \times n}$ be an arbitrary function for the input transformation. The $q$-layer $\mathsf{EGNN} : \mathbb{R}^{d \times n} \times \mathbb{R}^{3 \times n} \times \mathbb{R}^{3 \times 3} \to \mathbb{R}^{d \times n}$ can be defined as follows:*

$$\mathsf{EGNN}(A, F, L) := \mathsf{EGNN}_q \circ \mathsf{EGNN}_{q-1} \circ \cdots \circ \mathsf{EGNN}_1(\phi_{\mathrm{in}}(A), F, L).$$

**Remark 3.11.** *While functions $\phi_{\mathrm{msg}}$, $\phi_{\mathrm{upd}}$, and $\phi_{\mathrm{in}}$ are usually implemented as simple MLPs in practice, our theoretical result on equivariance and invariance works for any possible instantiation of these functions.*

### 3.3 CIRCUIT COMPLEXITY CLASS

In this section, we present Boolean circuits and key preliminaries for circuit complexity.

**Definition 3.12** (Boolean Circuit, implicit in page 102 on (Arora & Barak, 2009))**.** *Let $n \in \mathbb{Z}_+$. A Boolean circuit is defined as a directed acyclic graph (DAG) that realizes a function $C_n : \{0, 1\}^n \to \{0, 1\}$. The nodes of the graph are referred to as gates. Those with in-degree zero serve as input nodes, corresponding to the $n$ Boolean variables, while every other gate applies a Boolean function to the output of its predecessors.*

Since each circuit is limited to inputs of a fixed length, we rely on a sequence of circuits to handle languages that include strings of any length.

**Definition 3.13** (Circuit family recognizes languages, implicit in page 103 on (Arora & Barak, 2009))**.** *Consider a language $L \subseteq \{0, 1\}^*$ and a family of Boolean circuits $C = \{C_n\}_{n \in \mathbb{N}}$, we say that $C$ recognizes $L$ if every string $x \in \{0, 1\}^*$, $C_{|x|}(x) = 1 \iff x \in L$.*

By restricting the size and depth of circuit families, we can define certain complexity classes, for example $\mathsf{NC}^i$.

**Definition 3.14** ($\mathsf{NC}^i$, implicit in page 40 on (Arora & Barak, 2009))**.** *A language belongs to $\mathsf{NC}^i$ if it is decidable by a family of Boolean circuits of polynomial size $O(\mathrm{poly}(n))$, depth $O((\log n)^i)$, and built from AND, OR, and NOT gates with bounded fan-in.*

By extending AND and OR gates to unbounded fan-in, we obtain more expressive circuits, which define the class $\mathsf{AC}^i$.

**Definition 3.15** ($\mathsf{AC}^i$, (Arora & Barak, 2009)). *A language belongs to $\mathsf{AC}^i$ if it can be recognized by a family of Boolean circuits with polynomial size $O(\mathrm{poly}(n))$ and depth $O((\log n)^i)$, composed of* NOT, OR, *and* AND *gates, with* OR *and* AND *gates permitted unbounded fan-in.*

Since MAJORITY gates can simulate NOT, AND, and OR, an even larger class $\mathsf{TC}^i$ can be defined.

**Definition 3.16** ($\mathsf{TC}^i$, (Arora & Barak, 2009)). *A language belongs to $\mathsf{TC}^i$ if it can be recognized by a family of Boolean circuits of polynomial size $O(\mathrm{poly}(n))$ and depth $O((\log n)^i)$, composed of* NOT, OR, AND, *and* MAJORITY *gates with unbounded fan-in, where a* MAJORITY *gate outputs* 1 *when a majority of its inputs are active (1).*

**Remark 3.17.** *In Definition 3.16, the* MAJORITY *gates of $\mathsf{TC}^i$ may be replaced with* MOD *gates or* THRESHOLD *gates. Circuits that employ any of these gates are referred to as threshold circuits.*

**Definition 3.18** (P, implicit in page 27 on (Arora & Barak, 2009)). *A language belongs to* P *if it can be decided by a deterministic Turing machine in polynomial time.*

**Fact 3.19** (Hierarchy folklore, (Arora & Barak, 2009; Vollmer, 1999)). *The following class inclusions are valid for all $i \geq 0$:*

$$\mathsf{NC}^i \subseteq \mathsf{AC}^i \subseteq \mathsf{TC}^i \subseteq \mathsf{NC}^{i+1} \subseteq \mathsf{P}.$$

**Definition 3.20** (L-uniform, (Arora & Barak, 2009)). *A circuit family $C = \{C_n\}_{n \in \mathbb{N}}$ is said to be L-uniform if there exists a Turing machine that, given input $1^n$, outputs a description of $C_n$ using $O(\log n)$ space. A language $L$ belongs to a class such as L-uniform $\mathsf{NC}^i$ if it can be decided by an L-uniform circuit family $C_n$ satisfying the size and depth requirements of $\mathsf{NC}^i$.*

Next, we introduce a stronger notion of uniformity defined in terms of a time bound.

**Definition 3.21** (DLOGTIME-uniform). *A circuit family $C = \{C_n\}_{n \in \mathbb{N}}$ is DLOGTIME-uniform if there exists a Turing machine that, on input $1^n$, outputs a description of $C_n$ within $O(\log n)$ time. A language belongs to a DLOGTIME-uniform class if it can be decided by such a circuit family while also meeting the required size and depth bounds.*

The following lemmas characterize the depth and width of basic operations, which are essential in our study of circuit complexity. We first establish that fundamental floating-point operations can be implemented within $\mathsf{TC}^0$.

**Lemma 3.22** (Operations on floating point numbers in $\mathsf{TC}^0$, Lemma 10 and Lemma 11 of (Chiang, 2024)). *Assume the precision $p \leq \mathrm{poly}(n)$. Then we have:*

- *Part 1. Consider two $p$-bits float point numbers $x_1$ and $x_2$. As described in (Chiang, 2024), their addition, division, and multiplication can be carried out using a threshold circuit of polynomial size and constant depth $d_{std}$, which is DLOGTIME-uniform.*

- *Part 2. Given $n$ $p$-bits float point number $x_1, \ldots, x_n$, their iterated product can be simulated by a DLOGTIME-uniform threshold circuit of polynomial size and constant depth $d_\otimes$.*

- *Part 3. Given $n$ $p$-bits float point number $x_1, \ldots, x_n$, their iterated sum can be simulated by a DLOGTIME-uniform threshold circuit of polynomial size and constant depth $d_\oplus$. Note that a rounding step is applied after the summation.*

We now establish that the exponential function can also be approximated in $\mathsf{TC}^0$.

**Lemma 3.23** (Approximating the Exponential Operation in $\mathsf{TC}^0$, Lemma 12 of (Chiang, 2024)). *Assume the precision satisfies $p \leq \mathrm{poly}(n)$. For any $p$-bit floating-point number $x$, the function $\exp(x)$ can be approximated by a uniform threshold circuit of polynomial size and constant depth $d_{exp}$, achieving a relative error no greater than $2^{-p}$.*

Finally, we show that the square root function can be approximated within $\mathsf{TC}^0$.

**Lemma 3.24** (Approximating the Square Root Operation in $\mathsf{TC}^0$, Lemma 12 of (Chiang, 2024)). *Assume the precision satisfies $p \leq \mathrm{poly}(n)$. For any $p$-bit floating-point number $x$, the function $\sqrt{x}$ can be approximated by a uniform threshold circuit of polynomial size and constant depth $d_{sqrt}$, achieving a relative error no greater than $2^{-p}$.*

## 3.4 Floating Point Numbers

In this subsection, we present the basic definitions of floating-point numbers and their operations, which provide the computational framework for implementing GNNs on practical hardware.

**Definition 3.25** (Floating Point Numbers (FPNs), Definition 9 in (Chiang, 2024)). *A $p$-bit floating-point number* (FPN) *can be expressed as a pair of binary integers $\langle s, e \rangle$. Here, the significand $|s|$ takes values in $\{0\} \cup [2^{p-1}, 2^p)$, while the exponent $e$ lies within $[-2^p, 2^p - 1]$. The value of the* FPN *is calculated as $s \cdot 2^e$. When $e = 2^p$, the floating-point number represents positive or negative infinity, depending on the sign of $s$. We use $\mathbb{F}_p$ to denote the set of all the $p$-bit* FPN*s.*

**Definition 3.26** (Rounding, Definition 9 in (Chiang, 2024)). *Let $r \in \mathbb{R}$ be a real number with infinite precision. Its nearest $p$-bit representation is written as $\mathrm{round}_p(r) \in \mathbb{F}_p$. If two such representations are equally close, $\mathrm{round}_p(r)$ is defined as the one with an even significand.*

Then, we introduce the key floating-point operations used to compute the outputs of neural networks.

**Definition 3.27** (FPN operations, page 5 on (Chiang, 2024)). *Let $x$ and $y$ be two integers. We define the integer division operation $/\!/$ as follows:*

$$x /\!/ y := \begin{cases} x/y & \text{if } x/y \text{ is a multiple of } 1/4 \\ x/y + 1/8 & \text{otherwise.} \end{cases}$$

*Given two $p$-bits* FPN*s $\langle s_1, e_1 \rangle, \langle s_2, e_2 \rangle \in \mathbb{F}_p$, we define the fundamental operations on them as:*

$$\text{addition} : \langle s_1, e_1 \rangle + \langle s_2, e_2 \rangle := \begin{cases} \mathrm{round}_p(\langle s_1 + s_2 /\!/ 2^{e_1 - e_2}, e_1 \rangle) & \text{if } e_1 \geq e_2 \\ \mathrm{round}_p(\langle s_1 /\!/ 2^{e_2 - e_1} + s_2, e_2 \rangle) & \text{if } e_1 \leq e_2 \end{cases}$$

$$\text{multiplication} : \langle s_1, e_1 \rangle \times \langle s_2, e_2 \rangle := \mathrm{round}_p(\langle s_1 s_2, e_1 + e_2 \rangle)$$

$$\text{division} : \langle s_1, e_1 \rangle \div \langle s_2, e_2 \rangle := \mathrm{round}_p(\langle s_1 \cdot 2^{p-1} /\!/ s_2, e_1 - e_2 - p + 1 \rangle)$$

$$\text{comparison} : \langle s_1, e_1 \rangle \leq \langle s_2, e_2 \rangle \Leftrightarrow \begin{cases} s_1 \leq s_2 /\!/ 2^{e_1 - e_2} & \text{if } e_1 \geq e_2 \\ s_1 /\!/ 2^{e_2 - e_1} \leq s_2 & \text{if } e_1 \leq e_2. \end{cases}$$

Building on the previous definitions, we show that these basic operations can be efficiently executed in parallel using simple $\mathsf{TC}^0$ circuit constructions, as established in the following lemma:

**Lemma 3.28** (Computing FPN operations with $\mathsf{TC}^0$ circuits, Lemma 10 and Lemma 11 in (Chiang, 2024)). *Let $p$ be a positive integer representing the number of digits. If $p \leq \mathrm{poly}(n)$, then the following holds:*

- *Basic Operations: The operations "$+$", "$\times$", "$\div$", and comparison ($\leq$) between two $p$-bit* FPN*s, as defined in Definition 3.25, can be implemented by uniform threshold circuits of $O(1)$-depth and $\mathrm{poly}(n)$ size. Denote the maximum depth required for these basic operations as $d_{\mathrm{std}}$.*

- *Iterated Operations: The product of $n$ $p$-bit* FPN*s, as well as the sum of $n$ $p$-bit* FPN*s (with rounding applied after summation) can both be computed by uniform threshold circuits with $O(1)$-depth and $\mathrm{poly}(n)$ size. Let $d_{\otimes}$ and $d_{\oplus}$ denote the maximum circuit depth for multiplication and addition.*

In addition to the basic floating-point operations, some specialized operations can also be executed within $\mathsf{TC}^0$ circuits, as shown in the following lemmas:

**Lemma 3.29** (Computing $\exp$ with $\mathsf{TC}^0$ circuits, Lemma 12 in (Chiang, 2024)). *Let $x \in \mathbb{F}_p$ be a $p$-bit* FPN. *Provided that $p \leq \mathrm{poly}(n)$, there exists a uniform threshold circuit of $\mathrm{poly}(n)$ size and $O(1)$ depth that can approximate $\exp(x)$ with a relative error less than $2^{-p}$. We denote the maximum depth needed for this approximation by $d_{\exp}$.*

**Lemma 3.30** (Computing square root with $\mathsf{TC}^0$ circuits, Lemma 12 in (Chiang, 2024)). *Let $x \in \mathbb{F}_p$ be a $p$-bit* FPN. *If $p \leq \mathrm{poly}(n)$, then there exists a uniform threshold circuit of $O(1)$-depth and $\mathrm{poly}(n)$ size capable of computing $\sqrt{x}$ with relative error smaller than $2^{-p}$. Denote the maximum circuit depth required for this computation as $d_{\mathrm{sqrt}}$.*

**Lemma 3.31** (Computing matrix multiplication with $\mathsf{TC}^0$ circuits, Lemma 4.2 in (Chen et al., 2025a))**.** *Let $A \in \mathbb{F}_p^{n_1 \times n_2}$ and $B \in \mathbb{F}_p^{n_2 \times n_3}$ be two matrix operands. If $p \leq \mathrm{poly}(n)$ and $n_1, n_2, n_3 \leq n$, then there exists a uniform threshold circuit of $\mathrm{poly}(n)$ size, with maximum depth $(d_{\mathrm{std}} + d_\oplus)$ that can compute the matrix product $AB$.*

# 4 CIRCUIT COMPLEXITY OF CRYSTALLINE EGNNs

We first present the circuit complexity of basic EGNN building blocks in Section 4.1, and then show the circuit complexity for EGNN layers in Section 4.2.

## 4.1 CIRCUIT COMPLEXITY OF BASIC EGNN BUILDING BLOCKS

We begin by introducing a useful lemma that introduces the $\mathsf{TC}^0$ computation of trigonometric functions.

**Lemma 4.1** (Trigonometric function computation in $\mathsf{TC}^0$, Lemma 4.1 of (Chen et al., 2025a))**.** *Assume $p \leq \mathrm{poly}(n)$. For any $p$-bit floating-point number $x$, the function $\sin(x)$ and $\cos(x)$ can be approximated by a uniform threshold circuit of polynomial size and constant depth $8d_{\mathrm{std}} + d_\oplus + d_\otimes$, achieving a relative error no greater than $2^{-p}$.*

Then, we show that $k$-order Fourier Transforms, a fundamental building block for Crystalline EGNN layers, can be computed by the $\mathsf{TC}^0$ circuits.

**Lemma 4.2** ($k$-order Fourier Transform computation in $\mathsf{TC}^0$)**.** *Assume $p \leq \mathrm{poly}(n)$ and $k = O(n)$. For any $p$-bit floating-point number $x$, the function $\psi_{\mathrm{Ft},k}(x)$ defined in Definition 3.7 can be approximated by a uniform threshold circuit of polynomial size and constant depth $10d_{\mathrm{std}} + d_\oplus + d_\otimes$, achieving a relative error no greater than $2^{-p}$.*

*Proof.* According to Definition 3.7, for each $(i, j) \in [3] \times [k]$ there are two fixed cases:

**Case 1.** $j$ is even, then $Y_{i,j} := \sin(\pi j x_i)$. Computing $\pi j x_i$ uses $2d_{\mathrm{std}}$ depth and $\mathrm{poly}(n)$ size. Then, according to Lemma 4.1, we need to use $8d_{\mathrm{std}} + d_\oplus + d_\otimes$ and $\mathrm{poly}(n)$ size for the $\sin$ operation. Thus, the total depth of this case is $10d_{\mathrm{std}} + d_\oplus + d_\otimes$, and the size is $\mathrm{poly}(n)$.

**Case 2.** $j$ is odd, then $Y_{i,j} := \cos(\pi j x_i)$. Similar to case 1, the only difference is we need to use $\cos$ instead of $\sin$. According to Lemma 4.1, $\cos$ takes $8d_{\mathrm{std}} + d_\oplus + d_\otimes$ depth and $\mathrm{poly}(n)$ size, which is same as $\sin$ in case 1. Thus, the total depth of this case is $10d_{\mathrm{std}} + d_\oplus + d_\otimes$, and the size is $\mathrm{poly}(n)$.

Since all $[3] \times [k]$ elements in $Y$ can be computed in parallel, thus we need $3k$ parallel circuit with $10d_{\mathrm{std}} + d_\oplus + d_\otimes$ depth to simulate the computation of $Y$. Since $k = O(n)$, thus we can simulate the computation with circuit of $\mathrm{poly}(n)$ size and $10d_{\mathrm{std}} + d_\oplus + d_\otimes = O(1)$ depth. Thus $k$-order Fourier Transform can be simulated by a $\mathsf{TC}^0$ uniform threshold circuit. $\qquad\square$

We also show that MLPs are computable with uniform $\mathsf{TC}^0$ circuits.

**Lemma 4.3** (MLP computation in $\mathsf{TC}^0$, , Lemma 4.5 of (Chen et al., 2025a))**.** *Assume the precision $p \leq \mathrm{poly}(n)$. Then, we can use a size bounded by $\mathrm{poly}(n)$ and constant depth $2d_{\mathrm{std}} + d_\oplus$ uniform threshold circuit to simulate the MLP layer with $O(1)$ depth and $O(n)$ width, achieving a relative error no greater than $2^{-p}$.*

## 4.2 CIRCUIT COMPLEXITY OF EGNN LAYER

**Lemma 4.4** (Pairwise Message computation in $\mathsf{TC}^0$.)**.** *Assume $p \leq \mathrm{poly}(n)$, $d = O(n)$ and $k = O(n)$. Assume $\phi_{\mathrm{msg}}$ is instantiated with $O(1)$ depth and $O(n)$ width MLP. For any $p$-bit floating-point number $x$, the function $\mathsf{MSG}(F, L, H)$ defined in Definition 3.8 can be approximated by a uniform threshold circuit of polynomial size and constant depth $13d_{\mathrm{std}} + 2d_\oplus + d_\otimes$, achieving a relative error no greater than $2^{-p}$.*

*Proof.* We first analyze the arguments in for the $\phi_{\mathrm{msg}}$ function. The first two arguments do not involve computation. The third argument $L^\top L$ involves one matrix multiplication. According to

Lemma 3.31, we could compute the matrix multiplication using a circuit of $\text{poly}(n)$ size and $d_{\text{std}} + d_{\oplus}$ depth.

In order to analyze the last argument $\psi_{\text{FT},k}(f_i - f_j)$, we first analyze $f_i - f_j$, which takes $d_{\text{std}}$ depth and constant size. Then, according to Lemma 4.2, we can compute the $\psi_{\text{FT},k}(\cdot)$ with circuit of $\text{poly}(n)$ size and $10d_{\text{std}} + d_{\oplus} + d_{\otimes}$ depth. Therefore, we can compute the last argument $\psi_{\text{FT},k}(f_i - f_j)$ with circuit of $\text{poly}(n)$ size and $11d_{\text{std}} + d_{\oplus} + d_{\otimes}$ depth.

Next, since $d = O(n)$ and $k = O(n)$ according to Lemma 4.3, we can use circuit with $\text{poly}(n)$ size and $2d_{\text{std}} + d_{\oplus}$ to compute the $\phi_{\text{msg}}(\cdot)$ function.

Combining above, we can use circuit with $\text{poly}(n)$ size and $2d_{\text{std}} + d_{\oplus} + \max\{d_{\text{std}} + d_{\oplus}, 11d_{\text{std}} + d_{\oplus} + d_{\otimes}\} = 13d_{\text{std}} + 2d_{\oplus} + d_{\otimes} = O(1)$ depth to compute the pairwise message. Thus, pairwise message computation can be simulated by a $\text{TC}^0$ uniform threshold circuit. $\qquad\square$

**Lemma 4.5** (One EGNN layer approximation in $\text{TC}^0$, informal version of Lemma B.1). *Assume $p \leq \text{poly}(n)$, $d = O(n)$ and $k = O(n)$. Assume $\phi_{\text{msg}}$ and $\phi_{\text{upd}}$ are instantiated with $O(1)$ depth and $O(n)$ width MLPs. For any $p$-bit floating-point number $x$, the function $\text{EGNN}_i(A, F, H)$ defined in Definition 3.9 can be approximated by a uniform threshold circuit of polynomial size and constant depth $16d_{\text{std}} + 3d_{\oplus} + 2d_{\otimes}$, achieving a relative error no greater than $2^{-p}$.*

## 5 MAIN RESULTS

In this section, we present our main result, which show that under some assumptions, EGNN class in Definition 3.10 can be simulated by a uniform $\text{TC}^0$ circuit family.

**Theorem 5.1.** *If precision $p \leq \text{poly}(n)$, embedding size $d = O(n)$, the number of layers $q = O(1)$, $k = O(n)$, and all the functions $\phi_{\text{msg}}$, $\phi_{\text{upd}}$, and $\phi_{\text{in}}$ are instantiated with $O(1)$ depth and $O(n)$ width MLPs, then the equivariant graph neural network $\text{EGNN} : \mathbb{R}^{d \times n} \times \mathbb{R}^{3 \times n} \times \mathbb{R}^{3 \times 3} \to \mathbb{R}^{d \times n}$ which defined in Definition 3.10 can be simulated by the uniform $\text{TC}^0$ circuit family.*

*Proof.* Since $d = O(n)$, according to Lemma 4.3, the computation of first argument ($\phi_{\text{in}}(A)$) can be approximated by a circuit of $2d_{\text{std}} + d_{\oplus}$ depth and $\text{poly}(n)$ size. Last two arguments does not include computation.

Then, according to Lemma 4.5, for each EGNN layer, we need a circuit with $\text{poly}(n)$ size and $16d_{\text{std}} + 3d_{\oplus} + 2d_{\otimes}$ depth to simulate the computation.

Combining results above, since there are $q$ serial layer of EGNN, we need circuit of $\text{poly}(n)$ size and

$$q(16d_{\text{std}} + 3d_{\oplus} + 2d_{\otimes} + 2d_{\text{std}} + d_{\oplus}) = q(18d_{\text{std}} + 4d_{\oplus} + 2d_{\otimes})$$
$$= O(1)$$

depth to simulate the EGNN. Thus, the EGNN can be simulated by a $\text{TC}^0$ uniform threshold circuit. $\qquad\square$

## 6 CONCLUSION

We studied the computational expressiveness of equivariant graph neural networks (EGNNs) for crystalline-structure prediction through the lens of circuit complexity. Under realistic architectural and precision assumptions—polynomial precision, embedding width $d = O(n)$, $q = O(1)$ layers, and $O(1)$-depth, $O(n)$-width MLP instantiations of the message, update, and readout maps—we established that an EGNN as formalized in Definition 3.10 admits a simulation by a *uniform* $\text{TC}^0$ circuit family of polynomial size. Our constructive analysis further yields an explicit depth bound of $q(18d_{\text{std}} + 4d_{\oplus} + 2d_{\otimes})$, thereby placing a concrete ceiling on the computations performed by such models.

## ETHICS STATEMENT

This paper does not involve human subjects, personally identifiable data, or sensitive applications. We do not foresee direct ethical risks. We follow the ICLR Code of Ethics and affirm that all aspects of this research comply with the principles of fairness, transparency, and integrity.

## REPRODUCIBILITY STATEMENT

We ensure reproducibility of our theoretical results by including all formal assumptions, definitions, and complete proofs in the appendix. The main text states each theorem clearly and refers to the detailed proofs. No external data or software is required.

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

# Appendix

**Roadmap.** In Section A, we supplement the missing proofs in Section 3. In Section B, we show the missing proofs in Section 4.

## A    Missing Proofs in Section 3

**Fact A.1** (Equivalence of unit cell representations, formal version of Fact 3.5). *For any fractional unit cell representation $\mathcal{C}_{\mathrm{frac}} := (A, F, L)$ as Definition 3.4, there exists a unique corresponding non-fractional unit cell representation $\mathcal{C} := (A, X, L)$ as definition 3.1.*

*Proof.* **Part 1: Existence.** By Definition 3.1, we can conclude that $L$ is invertible since all the columns in $L$ are linearly independent. Thus, we can choose $X = L^{-1}F$ and finish the proof.

**Part 2: Uniqueness.** We show this by contradiction. First, we assume that there exist two different unit cell representations $\mathcal{C}_1 := (A, X_1, L)$ and $\mathcal{C}_2 := (A, X_2, L)$ for $\mathcal{C}_{\mathrm{frac}}$, i.e., $X_1 \neq X_2$. By Definition 3.3, we have $X_1 = X_2 = LF$, which contradicts $X_1 \neq X_2$. Thus, we finish the proof. $\qquad\square$

## B    Missing Proofs in Section 4

**Lemma B.1** (One EGNN layer approximation in $\mathsf{TC}^0$, formal version of Lemma 4.5). *Assume $p \leq \mathrm{poly}(n)$, $d = O(n)$ and $k = O(n)$. Assume $\phi_{\mathrm{msg}}$ and $\phi_{\mathrm{upd}}$ are instantiated with $O(1)$ depth and $O(n)$ width MLPs. For any $p$-bit floating-point number $x$, the function $\mathsf{EGNN}_i(A, F, H)$ defined in Definition 3.9 can be approximated by a uniform threshold circuit of polynomial size and constant depth $16d_{\mathrm{std}} + 3d_{\oplus} + 2d_{\otimes}$, achieving a relative error no greater than $2^{-p}$.*

*Proof.* We start with analyzing the arguments in $\phi_{\mathrm{upd}}(\cdot)$. The first argument does not involve computation. For the second argument, according to Lemma 4.4, we need circuit with $\mathrm{poly}(n)$ size and $13d_{\mathrm{std}} + 2d_{\oplus} + d_{\otimes}$ depth to simulate $\mathsf{MSG}_{i,j}(F, L, H)$ computation.

Then, for the summation $\sum_{j=1}^n \mathsf{MSG}_{i,j}(F, L, H)$, we can compute $n$ $\mathsf{MSG}_{i,j}(F, L, H)$ in parallel, and use a circuit with $d_{\oplus}$ width to perform the summation. Thus we can simulate the last argument with circuit of $poly(n)$ size $13d_{\mathrm{std}} + 2d_{\oplus} + 2d_{\otimes}$ depth to simulate the last argument.

Next, for $\phi_{\mathrm{upd}}(\cdot)$, since $d = O(n)$, according to Lemma 4.3, we can simulate $\phi_{\mathrm{upd}}(\cdot)$ with circuit of $\mathrm{poly}(n)$ size $2d_{\mathrm{std}} + d_{\oplus}$ depth. Finally, for the addition of $\mathbb{R}^d$ size vector, we need circuit $\mathrm{poly}(n)$ size and $d_{\mathrm{std}}$ depth to simulate it.

Combining circuits above, we can simulate $\mathsf{EGNN}_i(A, F, H)$ with a circuit of $\mathrm{poly}(n)$ size and

$$13d_{\mathrm{std}} + 2d_{\oplus} + 2d_{\otimes} + 2d_{\mathrm{std}} + d_{\oplus}d_{\mathrm{std}} = 16d_{\mathrm{std}} + 3d_{\oplus} + 2d_{\otimes}$$
$$= O(1)$$

depth to simulate the computation. Thus, one $\mathsf{EGNN}$ layer can be simulated by a $\mathsf{TC}^0$ uniform threshold circuit. $\qquad\square$

## LLM Usage Disclosure

LLMs were used only to polish language, such as grammar and wording. These models did not contribute to idea creation or writing, and the authors take full responsibility for this paper's content.

