# OpenReview forum: "Fundamental Limits of Crystalline Equivariant Graph Neural Networks: A Circuit Complexity Perspective"
_ICLR.cc/2026/Conference — Submitted to ICLR 2026_

### Official Review · Reviewer_bh4F · 2025-10-18

**Soundness:** 1
**Presentation:** 1
**Contribution:** 1
**Rating:** 2
**Confidence:** 5

**Summary:**

This paper proposes a method to analyze equivariant graph neural networks in the crystal domain using the circuit complexity perspective.

**Strengths:**

- Equivariant graph neural networks are a crucial research area in the field of crystals, and studying their expressive power is a crucial area of research.
- Using circuit complexity as an approach is a particularly interesting research point.
- This paper contains extensive theoretical content and analysis, reflecting the author's extensive research.

**Weaknesses:**

> **Motivation Concerns**

The current motivation of the paper relies on questionable assumptions regarding expressive power in graph learning.
- The analogy to the Weisfeiler–Lehman (WL) paradigm is not well grounded. WL is intended for topological graphs, where graph isomorphism does not have a known polynomial-time solution.
- In contrast, geometric graphs **do** admit polynomial-time graph isomorphism algorithms, and there already exists substantial work on complete geometric GNNs [A-C]. While these works rarely discuss periodic infinite graphs such as crystals, extending them to periodic settings is relatively straightforward.
- Moreover, the modeling of crystals using fractional coordinates implicitly ignores axis permutation equivariance/invariance. This simplification avoids dealing with symmetry-induced **systematic extinction**[D,E], a fundamental concept in crystallography. Without considering this, any claim of expressive power remains incomplete.
These issues leave the core motivation unconvincing.

> **Related Work Scope**

The inclusion of topics such as CSP and Flow Matching is not clearly justified. Their relevance to the main problem is unclear and should either be clarified or removed.
> **Preliminaries vs. Contributions**

A substantial portion of the paper is devoted to common definitions and lemmas.
- For non-experts, this makes the paper difficult to follow.
- For experts, these are standard facts and do not add value.
- More importantly, it becomes unclear which parts constitute the actual contributions of the authors.
For instance, it is debatable whether Definition 3.10 should be positioned as a contribution—it could be placed in preliminaries without loss.

> **Lack of Empirical Evidence**

No experimental validation is provided. If the theoretical bounds are intended to have practical implications, comparison with prior works such as eSEN [F] is necessary to demonstrate relevance.

> **Writing and Presentation**

The exposition could be significantly improved. The absence of visual illustrations makes it challenging to grasp the intuition behind the framework.

> **Suspected Plagiarism**

I would appreciate a clear, verifiable clarification in your rebuttal regarding potential near-verbatim overlap with another submission. The Definitions and Lemmas in Section 3.3 (Lines 251–323) appear near-verbatim—nearly character-by-character—with Appendix C.2 (Lines 897–976) of ICLR 2026 submission #2462 (https://openreview.net/forum?id=rj7PF436kF) "Towards High-Order Mean Flow Generative Models: Feasibility, Expressivity, and Provably Efficient Criteria," showing line-by-line correspondence in statements, notation, and result ordering, with only minimal paraphrasing.

Please provide dated provenance for Section 3.3, a precise statement of its relationship to the cited submission (including any dependence or text/result reuse with appropriate citation), identification of any substantive differences or an explicit confirmation if none exist, and disclosure of any related manuscripts to confirm compliance with the venue’s originality policies.

> **Reference**

[A] On the Expressive Power of Geometric Graph Neural Networks

[B] Universally Invariant Learning in Equivariant GNNs

[C] On the Completeness of Invariant Geometric Deep Learning Models

[D] Point group analysis in particle simulation data

[E] Are High-Degree Representations Really Unnecessary in Equivarinat Graph Neural Networks?

[F] Learning Smooth and Expressive Interatomic Potentials for Physical Property Prediction

**Questions:**

See Weakness.

---

### Official Review · Reviewer_cSkq · 2025-10-23

**Soundness:** 1
**Presentation:** 1
**Contribution:** 2
**Rating:** 2
**Confidence:** 4

**Summary:**

This paper demonstrates, from the perspective of circuit complexity, that the EGNN model can be simulated by the $\mathrm{TC}^0$ family under certain regularity conditions for crystal-related tasks, thereby establishing a lower bound on the circuit complexity of the EGNN model.

**Strengths:**

The idea of considering the expressive power of graph neural networks from the angle of circuit complexity is novel. Few other papers studying network expressivity have discussed issues related to floating-point errors in computation, whereas this paper does.

**Weaknesses:**

1. Significant room for improvement in writing. There is a large degree of redundancy in the related work and preliminaries sections, which severely hinders reader comprehension. In the related work, "Flow Matching" is a topic with little relevance to the main theme, yet the author dedicates significant space to describing it without any subsequent analysis in the rest of the paper. Furthermore, "Geometric Deep Learning" is a very broad topic, but the author only discusses crystal-related content; therefore, it should not be listed as a preliminary topic with extensive descriptions of other works. In the preliminaries, Section 3.1 defines several similar representations for crystals, but in practice, the author only needs to detail the specific "unit fractional representation" that is used. The author urgently needs to condense these two sections.

2. The core motivation is not adequately justified. The main text includes descriptions of the Weisfeiler-Lehman (WL) test, a topic likely familiar to readers interested in the expressive power of GNNs and equivariant networks. The author claims, “These differences make such results ill-suited,” implying that WL test results are not suitable for crystal tasks. However, I did not understand the problem with the WL test from the introduction, nor did the author mention it again later. I believe the author should clearly explain these points that are of interest to the reader.

3. Core concepts are undefined. This section needs to be supplemented with definitions and motivations. The majority of the paper presents abstract theory without any diagrams or necessary descriptions, creating a significant barrier to reading. Section 3.3 is a major problem area. At the beginning of the definitions, the author introduces Boolean circuits as a core concept, but there is no definition for this circuit in the paper. I do not know why this concept is introduced or how it connects to EGNNs. The transition from the previous section seems very abrupt.

4. Lack of necessary extended discussion. The paper discusses the expressive power of EGNNs on crystal data, but after reading it, besides an abstract conclusion, the questions readers care about remain unanswered. For example, what representational flaws in existing models does this conclusion reveal? How can models be improved to resolve these flaws?

5. Ambiguity in the statement of contributions. One of the contributions listed in the introduction is "Formalizing EGNNs’ structure." This description is vague. The structure of EGNNs was already given in the original paper and cannot be a core contribution. I suspect the author may have intended to say they are modeling the structure of EGNNs specifically for crystal-related tasks. The author should emphasize this point.

This paper is not suitable for acceptance. A competent paper should focus on making the author's contributions understandable to the reader, not on piling up theories and proofs that obstruct comprehension. However, this paper begins with a series of abstract definitions, and its presentation does little to improve the reader's cognitive understanding of EGNN's expressive power.

**Questions:**

See weaknesses.

---

### Official Review · Reviewer_UNba · 2025-11-01

**Soundness:** 3
**Presentation:** 3
**Contribution:** 2
**Rating:** 4
**Confidence:** 3

**Summary:**

The paper studies the theoretical expressiveness of crystalline equivariant graph neural networks from the computational complexity pespective. The main result proves that these equivariant models can be simulated by $TC^0$ circuits, providing an upper bound of the computational power.

**Strengths:**

- The paper is the first to provide a rigorous circuit-complexity analysis of equivariant graph neural networks (EGNNs) in the context of crystalline structures, offering a new theoretical perspective that bridges geometric deep learning and computational complexity theory.

- The paper is well organized and clearly written. The results read technically sound, although I do not check the proofs in details.

**Weaknesses:**

The literature on circuit-complexity bounds for Transformers provides valuable insights into the types of functions that cannot be simulated by the architectures, such as constant-depth, log-precision transformers cannot perform arithmetic operations [1]. These results are closely related to the reasoning capabilities that people are interested in. I am curious, in the context of crystalline equivariant graph neural networks, what conceptual or practical insights the paper’s main result offers, specifically, what classes of physical or structural reasoning tasks could fall beyond the expressive power of EGNNs as characterized by the $TC^0$ bound. My concern is that physically meaningful computation is in continous domain, while circuit complexity may not be a good fit.


I would be happy to raise my score if the authors can adequately address these concerns.




[1] Feng, Guhao, et al. "Towards revealing the mystery behind chain of thought: a theoretical perspective." Advances in Neural Information Processing Systems 36 (2023): 70757-70798.

**Questions:**

See Weaknesses.

---

### Official Review · Reviewer_cbf5 · 2025-11-11

**Soundness:** 3
**Presentation:** 2
**Contribution:** 2
**Rating:** 4
**Confidence:** 3

**Summary:**

This paper studies the computational complexity of equivariant graph neural networks (EGNN) for crystalline-materials. Based on existing theory of circuit complexity, especially previous work on circuit complexity of transformers [Chiang 2024, Chen et al. 2025] a theory for circuit complexity of EGNNs is derived, especially for the case that the EGNN computes messages from fractional atom coordinates of a crystal lattice via the k-order Fourier transform.

**Strengths:**

The paper transfers existing theory that was mainly derived for transformers to graph neural networks. Additional theory for equivariant graph neural networks on crystal lattices from [Jiao et al. 2023] is used to transfer this theory to EGNNs on crystal lattices. The main contributions of this work are the formal definition of a EGNN layer that acts not only on hidden representations but also on fractional coordinates of atoms in the lattice. Then these results are combined to derive the main theorem.

**Weaknesses:**

A big part of the text are Lemmas taken from previous work. While it is great that the authors made the work self-contained, some of these Lemmas could have been moved to the appendix to provide further details on the Lemmas and the Theorem contributed in this work. The definition of the part of the message passing function, acting on fractal coordinates via the k-order Fourier transform appears like a restriction, for which further details should be discussed further in the text (see also questions below).

Technical remarks:

Chiang 2024, please cite the TMLR paper.

**Questions:**

Q1. Does the theory also apply to EGNNs which do not make use of the Fourier transform? (e.g. by defining appropriate boundary conditions  for the unit cell, as e.g. in [1]).

Q2. Does the formulation of the EGNN layer's function, that acts on fractional coordinates, by the k-order Fourier transform result in a restriction of the model's expressivity?

Q3. Does the formulation using the k-order Fourier transform result in a restriction of the applicability of the presented theory?

Q4. The Fourier transform (with sin/cos basis) assumes infinite support. How does this relate to models with local support?

[1] Ruff, R., Reiser, P., Stühmer, J., & Friederich, P. (2024). Connectivity optimized nested line graph networks for crystal structures. Digital Discovery, 3(3), 594-601.

**Details Of Ethics Concerns:**

As pointed out by reviewer bh4F, this paper contains sections which are potentially (self-)plagiarism or (in parts) a dual submission.

---

### Meta-Review · Area_Chair_eMn6 · 2025-12-29

**Summary:**

The reviewers identified several concerns that were not addressed during the rebuttal. These include issues with writing quality and clarity of presentation, limited applicability, unclear motivation, lack of empirical validation, and concerns regarding potential plagiarism. In light of these issues, I recommend rejection.

**Reviewer Concerns:**

The authors did not participate in the discussion phase. As a result, several concerns raised by the reviewers are unaddressed, including issues with writing quality and clarity of presentation (``cbf5``, ``cSkq``, ``bh4F``), limited applicability of the theoretical results (``cbf5``, ``UNba``), insufficient motivation for the proposed analysis (``cSkq``, ``bh4F``), lack of empirical validation (``bh4F``), and concerns regarding potential plagiarism (``bh4F``).

**Reviewer Scores:**

All reviewers are leaning towards rejection (ratings 2 or 4). Since the authors did not participate in the author–reviewer discussion, no change in the reviewers’ scores would be expected.

---

### Decision · Program_Chairs · 2026-01-26

Reject